# Short-Term Outcomes of an ESDM Intervention in Italian Children with Autism Spectrum Disorder following the COVID-19 Lockdown

**DOI:** 10.3390/children10040691

**Published:** 2023-04-06

**Authors:** Raffaella Devescovi, Giulia Bresciani, Vissia Colonna, Marco Carrozzi, Andrea Dissegna, Maria Antonella Celea, Devid Cescon, Sara Frisari, Marcella Guerrieri, Francesca Placer, Monica Stocchi, Chiara Terpini, Costanza Colombi

**Affiliations:** 1Division of Child Neurology and Psychiatry, Institute for Maternal and Child Health—IRCCS “Burlo Garofolo”, 34137 Trieste, Italy; 2Department of Life Sciences, University of Trieste, 34127 Trieste, Italy; 3CIMeC Centre for Mind/Brain Sciences, University of Trento, 38068 Rovereto, Italy; 4Unit of Neurodevelopmental Disorders and Developmental Psychopathology, Azienda Sanitaria Universitaria Giuliano Isontina, 34132 Trieste, Italy; 5IRCCS Fondazione Stella Maris, 56128 Calambrone, Italy

**Keywords:** autism, COVID-19, ESDM, parents, challenging behavior

## Abstract

The COVID-19 pandemic caused a temporary lockdown period in Italy, during which the delivery of in-person treatment for children with autism spectrum disorder (ASD) in public health services was discontinued. This occurrence represented a crucial challenge for both families and professionals. We assessed the short-term outcomes of a sample of 18 children who received an early intervention with the Early Start Denver Model (ESDM), delivered at low intensity over one year in the pre-pandemic period, after six months of interruption of in-presence treatment due to lockdown restrictions. Children who received the ESDM treatment maintained their gains in sociocommunicative skills and did not exhibit any developmental regression. Additionally, there was evidence of a decrease in the restrictive and repetitive behavior (RRB) domain. The parents, who were already familiar with the principles of the ESDM, only received telehealth support from therapists that aimed to sustain the gains already achieved. We believe that it is always helpful to support parents in their daily lives by implementing interactional and play skills with their children to integrate and consolidate the results obtained in the individual interventions conducted by experienced therapists.

## 1. Introduction

One of the neurodevelopmental disorders that occur frequently is autism spectrum disorder (ASD). ASD comprises impairments in social interaction and communication, along with repetitive and stereotyped behaviors [1]. The Centers for Disease Control and Prevention (CDC) reported in 2020 that 1 in 54 children under the age of 8 in the United States have been diagnosed with autism spectrum disorder (ASD) [2].

A growing body of research evidence supports the positive impact of early interventions on the developmental trajectories of autistic children. Naturalistic developmental behavioral interventions (NDBIs) are evidence-based interventions that merge the principles and strategies of applied behavior analysis (ABA) with developmental science [3]. The ESDM is an evidence-based manualized NDBI for children aged between 12 and 60 months that implements behavioral teaching strategies within the context of naturalistic developmentally appropriate activities [3,4]. The ESDM is an individualized intervention comprising a developmentally based curriculum and specific teaching strategies. Before the beginning of the intervention, a trained therapist established individualized learning objectives for each child based on an assessment conducted via the use of the ESDM curriculum checklist [5]. Learning objectives are established for each developmental area, including communication, motor skills, cognitive development, social interaction, imitation, play, and daily living skills. The ESDM can be implemented within a variety of contexts, including individually delivered specialized intervention and school groups, parent coaching, and by a variety of professionals, including psychologists, behavioral analysts, speech therapists, and occupational therapists. Many recent studies have documented the effectiveness of the ESDM for toddlers with, or at risk for, ASD. Fuller [6], in a recent meta-analysis, described 12 studies comprising 640 autistic children (286 intervention and 354 control). Such a meta-analysis demonstrated the efficacy of the ESDM in improving language and cognitive outcomes among young autistic children. The analysis revealed a moderate significant overall effect size, indicating that the ESDM is a promising intervention for improving areas impacted by the early onset of ASD. It is also important to mention that no significant effect sizes were observed for other measures, and concerns were raised from authors about the scientific rigor of many of the included studies. Moreover, Gao et al. [7] and Wang et al. [8], in their studies, aimed to investigate the impact of the ESDM on children with ASD. Gao et al.’s results showed that ESDM training also improved the core symptoms of autistic children and relieved parenting stress. On the other hand, Wang et al. found that the ESDM intervention led to a moderate improvement in cognition, autism symptoms, and language domains. The authors in the meta-analysis also found that the impact of the intervention on autism symptoms and language differed between Asian and Western countries, with larger effect sizes seen in Asian countries.

Devescovi and colleagues [9] conducted community studies in Italy to implement and assess the effectiveness of the ESDM. Their findings showed that a cohort of 21 autistic children, aged between 20 and 36 months, who received 3 h of a weekly ESDM intervention for 15 months, demonstrated notable progress in their cognitive abilities as well as language skills. Colombi et al. [10] evaluated the outcomes of 22 young autistic children who received an ESDM intervention in a center-based setting for 6 h per week for 6 months, compared with 70 children who received treatment as usual (TAU). The authors found that, after 3 months and 6 months of treatment, the children in the ESDM group improved more in cognitive, verbal, social, and adaptive skills. In their study, Contaldo and colleagues [11] observed 32 children with autism who underwent a community-based ESDM intervention for one year. They identified various predictors of treatment response, such as nonverbal skills prior to treatment, severity of symptoms, range of actions as well as gestures, and lexical comprehension. The ESDM intervention consisted of two types of sessions for each child: a 3 h group treatment session, comprising 2 h of group treatment and 1 h of individual treatment, followed by a separate 1 h individual session. The developmental progress of toddlers with ASD was examined by Cucinotta and colleagues [12], who investigated the effects of three distinct types of early interventions. The study included 90 children, who were classified into 3 groups, with 36 receiving treatment as usual, 13 receiving an early intensive behavioral intervention (EIBI), and 41 receiving an ESDM intervention. The three groups participated in their interventions for one year, at the same intensity of approximately 6 h per week. The results of the study showed that all three interventions improved the developmental profiles of the participants. The significance of an early intervention based on naturalistic developmental behavioral principles, including the Early Start Denver Model, is highlighted by the authors. Finally, an early detection and intervention regional program was conducted, named S.F.I.D.A. (Screening, Friuli Venezia Giulia, Intervention, Diagnosis, Autism). The project, carried out within the public health system of a region in north-eastern Italy, compared two groups of children treated with the ESDM and TAU implemented at a very low intensity (2 h per week) over the course of 1 year, investigating the feasibility and effectiveness of the ESDM provided in community settings. The results [13] showed that it is feasible to deliver the ESDM intervention within the Italian public health system. Moreover, the ESDM groups, in comparison to the TAU group, showed significantly larger improvements in cognitive, language, adaptive, and social communication skills. Parental satisfaction with the intervention was also high. The authors concluded that the ESDM is a feasible and effective early intervention for young autistic children delivered within the public services.

The current study represents the continuation of the S.F.I.D.A. project, which aims to investigate the short-term outcomes in the ESDM-treated group after the discontinuation of the treatment, conventionally delivered by therapists, for a period of six months due to the COVID-19 pandemic.

Italy was one of the first European countries to register cases of COVID-19, and was second to China in the number of cases. Following the declaration of a national emergency by the Italian government, social distancing measures were implemented, which included the closure of schools and of many commercial activities in the northern regions. These measures were eventually extended to the rest of the country. All schools were closed and economic activities were restricted in March 2020, following the Italian government’s declaration of a state of national emergency, as per a presidential decree [14]. Degli Espinosa et al. [15] found that the lockdown period, which was less restrictive and expected to last for a couple of weeks, resulted in an increase in challenging behavior and some skill losses in autistic children. Indeed, the COVID-19 outbreak exacerbated daily challenges, especially for autistic children. Shifting routines is a well-documented substantial issue for autistic children [16,17]. The pandemic meant that the home environment became the main intervention setting for autistic children and their families, compelling parents and therapists to adopt intervention strategies that had never or rarely been tried earlier, such as parent-mediated interventions with remote therapist supervision.

Our aim was, therefore, to investigate whether the improvements acquired by the children after one year of continuous in-person treatment with the ESDM had been maintained, despite its withdrawal for six months due to the pandemic restrictions.

## 2. Materials and Methods

### 2.1. Design

This study is a continuation of the quasi-experimental controlled treatment investigation that was carried out as part of the S.F.I.D.A project. The project involved implementing the Early Start Denver Model (ESDM) within the public health system of the Friuli Venezia Giulia (FVG) region in Italy. The study was funded by the FVG region under Regional Law 17/2014, which supports research in the clinical, translational, basic, epidemiological, and organizational domains. The Single Regional Ethics Committee (CEUR-2017-PR-050-BURLO) approved the study. The children who received ESDM treatment were compared to a concurrent non-randomized control group undergoing treatment as usual (TAU). Assessments were conducted at baseline (T0), after six months (T1), and after 12 months (T2) of the intervention. Prior to the onset of the lockdown in March 2020, the ESDM-treated group, comprising 19 children, underwent two one-hour sessions of individual ESDM treatment per week for 12 months, which were delivered in a center-based setting by therapists who had received ESDM training and fidelity certification from a certified trainer, as per the procedures outlined in Dawson and Rogers [18]. A detailed description of the study can be found in Devescovi et al. [13]. During the lockdown, due to pandemic restrictions, the in-person intervention was no longer deliverable for nearly 6 months and replaced by parent-mediated intervention supervised remotely by therapists when this was possible, but without a manualized and homogeneous procedure for the whole group. Families were supported with materials and instructions via remote connection, on average for two hours per week, but not all of the parents could put into practice what they had so far primarily observed therapists carrying out with their children. In our study, parents were coached by their children’s therapists in the skills necessary for managing several activities during the in-person treatment sessions and becoming able to develop interactional and play skills with their children based on ESDM principles, without having been exposed to training in a rigorous coaching method.

Before restarting the in-person treatments, the children were evaluated to verify the maintenance of the acquired skills (T3) in order to ascertain whether the improvements achieved at T2 had been maintained or whether regression had occurred after those six months of treatment interruption. The workflow of the study is represented in Figure 1.

In the present study, we did not include the control group that was present in previous work [13]. This decision was influenced by the impact of the pandemic on regional public services in addition to the depletion of specialized clinical staff dedicated to assessment but also blind to the intervention. 

### 2.2. Participants

The group study included 18 out of the 19 children (17 boys) aged 19–43 months (mean age of 29.5 months), because one family was refused access to the hospital due to the pandemic. The sociodemographic characteristics of the sample are described in Table 1 and Table 2.

The inclusion criteria were as follows: (a)Experienced clinicians administered the Autism Diagnostic Observation Schedule—Second Edition (ADOS-2) to assess the diagnosis of ASD as per the DSM-5 criteria;(b)The parents’ consent to participate in the 12-month intervention and to continue with post-pandemic follow-up was required;(c)Hearing and vision screened within a normal range;(d)Ability to use hands and ambulate.

The exclusion criteria for the study were as follows: (a) documented neurological disorders, such as epilepsy, and (b) notable sensory or motor impairments, such as cerebral palsy.

### 2.3. Measures

We have only partially adopted the assessment protocol used in the previous study [13]. In detail, we could not repeat the assessment of intellectual functioning because, according to the WPPSI-III manual, it was reported that a training effect could be produced, especially for the performance subtests, for which it was recommended to wait one to two years [19,20]. In addition, we did not administer the ESDM curriculum checklist [5] since the children examined had not resumed in-person ESDM treatment.

#### 2.3.1. Autism Diagnostic Observation Schedule—Second Edition (ADOS-2)

The ADOS-2 [21,22], which is a standardized diagnostic observational instrument, was used to evaluate autism symptoms related to social affect (SA), play, and restricted and repetitive behaviors (RRBs). To compare the severity of ASD, the calibrated severity scores overall (CSS overall) were used along with CSS scores for the restricted and repetitive behavior algorithm (CSS-RRB) and the social affect algorithm (CSS SA). These scores were calculated following the procedures outlined in Hus et al. [23] and Esler et al. [24].

#### 2.3.2. Vineland Adaptive Behavior Scales—Second Edition (VABS-II)

The VABS-II [25,26] is a standardized structured parent interview that assesses adaptive behavior from birth to adulthood. Communication, daily living, socialization, and motor skills are the four domains into which the VABS-II’s subscales are divided: standard scores, age equivalents, and an adaptive behavior composite (ABC).

#### 2.3.3. Behavior Observation of Social Communication Change (BOSCC)

The BOSCC [27] is a treatment response measure designed to track changes in minimally verbal social communication habits. Western Psychological Services and Dr Catherine Lord permitted us to use a measure preliminary version (August 2016). The BOSCC was created by modifying and expanding ADOS-2 [21] codes. Higher scores reflected more abnormal conduct. Items were coded on a 6-point scale from 0 to 5, with higher scores reflecting more atypical behaviors. It consisted of 12 items with a total score, and included scores in social communication (SC) as well as restricted and repetitive behaviors (RRBs). We created the BOSCC through the use of 15-min films of parent–child interaction with the same toys in different dyads and at various times. We classified the 10 min acquired by removing the initial and last 2 as well as 12 min from the 15-min recordings. The total, the SC and RRB scores were calculated by adding and averaging the scores from the two 5-min periods of each parent–child interaction. All interactions between the child and caregiver (mother or father) were recorded. One of the developers who helped create the measure conducted the training, having previously undergone BOSCC training at the Center for Autism and the Developing Brain (CADB). For the current study, three research assistants holding master’s degrees coded the videos after establishing reliability based on the authors’ criteria, which required scores within 1 point for over 80% of the questions in a given segment and an overall score within 3 points. Each coder was required to meet both criteria for 6 consecutive 5-min segments.

### 2.4. Statistical Analysis

We compared children’s scores at two time points (T2 and T3) via the use of the Wilcoxon signed-rank test. We also reported effect sizes (*r* = z/√N). The analysis of the data was performed in R [28].

## 3. Results

Table 3 illustrates the median results for each measure at T2 and T3, including the statistical analysis from the Wilcoxon signed-rank test. As observed, the children’s scores were relatively stable across most of the measures, except for restricted and repetitive behaviors assessed by the ADOS (CSS-RRB) and BOSCC (RRBs). Both questionnaires produced consistent results, indicating a decrease in restricted and repetitive behaviors from T2 to T3 (ADOS CSS-RRB: Mdn. of T2 = 8, Mdn. of T3 = 7, *p* = 0.044, r = 0.47; BOSCC RRB: Mdn. of T2 = 5, Mdn. of T3 = 3, *p* = 0.028, r = 0.52).

The statistically significant results of the CSS-ADOS-2 are given as follows:CSS-RRB: The ESDM group’s scores changed significantly from T2 to T3 (Mdn. of T2 = 8), (Mdn. of T3 = 7), *p* = 0.044, r = 0.47.

The statistically significant results of the BOSCC test are reported below:RRBs: The ESDM group’s scores changed significantly from T2 to T3 (Mdn. of T2 = 5), (Mdn. of T3 = 3), *p* = 0.028, r = 0.52.

All results are shown in Table 3. In Table 4, we have reported the means and ranges of the scores obtained from the tests used.

## 4. Discussion

The current study examined the impact of an early intervention based on the ESDM, delivered at a low intensity between the ages of 19 and 43 months (mean age of 29.5 months) for one year, after six months of the discontinuation of in-person treatment due to the pandemic. We observed that children treated with the ESDM maintained the gains achieved in sociocommunicative abilities and did not show developmental regression. Moreover, there was evidence for a reduction in severity in the restrictive and repetitive behavior (RRB) domain, revealed in the assessment conducted by expert clinicians unfamiliar with the intervention.

In our previous study [13], we compared the effects of the ESDM intervention with those of TAU provided within the Italian public services for autistic children. The experimental group in this study received the ESDM intervention and was the same group of children studied in the present research. Our findings showed that the children in the experimental group receiving the ESDM intervention showed more significant improvements in the communication domain, including expressive and receptive language, as well as in social and imitation skills; however, we did not observe any improvement in the restrictive and repetitive behaviors (RRB) domain. As reported by Berry et al. [29], despite its significance, research in the field of repetitive and restricted behaviors (RRBs) in ASD has been limited; however, in recent years, there has been a rise in research exploring RRBs in ASD from various perspectives, such as developmental psychology, neurobiology, psychiatry, and others. Although this may be the case, the lack of integration among these disciplines has hindered the formation of a comprehensive understanding of RRBs in ASD. Rogers et al. [30] conducted a study that compared an ESDM group to a TAU group; the ESDM group received a comprehensive, early intervention program delivered for 20 h per week over two years, focusing on both social communication skills and RRBs. The study showed that the ESDM group improved more significantly in cognitive, language, adaptive, and social communication skills compared to the TAU group after the intervention, and these improvements were maintained at the 12-month follow-up assessment.

A follow-up study was conducted by Estes and colleagues [31] to investigate the long-term effects of an early intervention on 39 children who were enrolled in a randomized clinical trial at 18–30 months of age and received a high-intensity intervention for a duration of 2 years. They highlighted that the ESDM group maintained the gains achieved post-treatment through overall developmental domains, including maladaptive and repetitive behaviors. Furthermore, the authors noted that there was no evidence of these children experiencing a setback in development, skill loss, or slowing progress on standardized tests after the cessation of early intensive services.

Interestingly, in our study, we observed a reduction in RRBs from T2 to T3. Given that RRBs represent an important area of impairment in ASD and yet are scarcely targeted in early interventions for autistic children, the discussion and further evaluation of such a finding seems noteworthy. The ESDM does not directly target RRBs; however, we hypothesize that several ESDM strategies might have led to this outcome. Firstly, the ESDM specifically promotes social orientation and attention by implementing strategies such as positioning in front of the child and imitating the play partner; thus, it is possible that increased attention on others leads to expanded play schemes by imitating the play partner. Secondly, the ESDM supports flexibility through several strategies, including elaboration and transitioning. During the elaboration phase of each ESDM joint activity, the therapist participates in the activity chosen by the child by demonstrating a variety of new actions. Moreover, during each ESDM session, the therapist and child perform many transitions by closing an activity and starting a new one as well as by moving through different play corners, such as the floor, a table, a couch, and a beanbag. Finally, by fully including parents in the therapy session it is likely that flexibility continues to be practiced naturally throughout the day in many different environments. Our hypothesis explaining a reduction in RRBs as an outcome of the ESDM should be further tested; however, it is credible that the strategies previously described promote flexibility and the acquisition of new play schemes, thus making REBs less salient for a child. Rather than directly targeting a decrease in RRBs, the ESDM may indirectly reduce them by expanding the play repertoire of a child. This might also explain why a reduction in RRBs does not occur in the short term at T2, but is observable later.

We are aware that our study presents some limitations, including the small size of the sample and the lack of a control group. Unfortunately, due to the pandemic the clinical staff, dedicated to research and unfamiliar with interventions, were no longer allowed access to public services, so it was not possible to repeat the assessments. Another limitation was that, during the time of the lockdown, parent-mediated interventions were implemented with remote supervision from therapists, but this approach was not standardized for all families, as some parents were unable to effectively replicate the techniques previously demonstrated by therapists with their children. Moreover, parents in the pre-pandemic period had not received fidelity coding from therapists. The P-ESDM fidelity tool, according to Rogers and Dawson [32], is a 5-point Likert-based rating system that assesses the degree to which parent behaviors align with the child-centered, responsive interactive style utilized in the ESDM. It is used to document the implementation rate of the technique. Despite these limitations, the findings from this study add to the growing body of evidence, suggesting that the ESDM is a promising intervention for young autistic children and that gains are maintained even after the termination of the intervention.

We maintain that one of its strengths is the emphasis on the importance of parental involvement in maintaining the gains achieved by children during the COVID-19 restrictions. We strongly believe that the coaching in the one-year treatment period before the pandemic made parents more receptive to the implementation of the advice that therapists could provide remotely during the lockdown. Indeed, as reported by Rogers [32], including parents and other family members in early intervention programs for autism have been shown to be important for their efficacy. Siracusano et al. [33] also reported that parents who received online support during the lockdown demonstrated significant improvement in some domains that evaluated self-care, safety, home life, and environmental care. They assert that this finding could be explained by the potential role of online training in implementing parental strategies to improve practical skills at home. Furthermore, the authors emphasized the positive effects of parental presence at home and time spent with their children by establishing daily stable routines. We can hypothesize that these changes in daily life, in addition to the reduction in unpredictable events, may have contributed to the reduced severity of stereotypic and repetitive behaviors. Finally, we believe that offering the opportunity for parents to receive the treatment at a fidelity level would be beneficial and necessary, as parents play a crucial role in ensuring that the treatment is generalized in the home and everyday environment in the future.

In summary, we observed that children who received the ESDM intervention during the period leading up to the pandemic through individual sessions did not show any developmental regression despite a six-month treatment interruption period due to the lockdown. We believe that the support that parents were able to give their children by availing themselves of supervision provided only through telehealth was crucial. Indeed, as highlighted by Rogers [32], the effects of parent-delivered toddler interventions are likely mediated by the quality and quantity of parent–child interactions. Additionally, it is possible that the lockdown situation provided an opportunity for parents, who are often occupied with their work, which takes them out of the home, to spend more time with their children, thereby enhancing the quality and frequency of interactions between them. We believe that the implementation of parent-mediated intervention at times when in-person treatment is not possible, such as during the lockdown, could be repeated in the future and help to reinforce treatment conducted in-person. This experience also reinforces the hypothesis that the direct involvement of parents promotes the generalization of the skills learned by children, resulting in a more comprehensive and successful intervention approach.

## 5. Conclusions

In conclusion, the present study analyzed the impact of a one-year early intervention program based on the Early Start Denver Model (ESDM) for young autistic children who underwent a pandemic-forced stop for 6 months. Our results also showed sustained positive outcomes of the intervention in areas that do not seem to improve immediately after treatment, such as RRBs.

The study also highlighted the importance of parental involvement, suggesting the possibility of telehealth interventions for those families who have difficulty physically accessing public services. This finding adds to the growing body of evidence supporting the ESDM as a promising intervention for autistic children. Further research is needed to validate the efficacy and effectiveness of the ESDM with larger samples and longer follow-up periods.

## Figures and Tables

**Figure 1 children-10-00691-f001:**
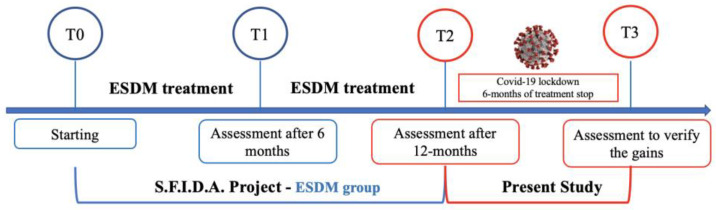
The workflow of the study. T0 baseline, T1 after six months, T2 after 12 months of the intervention and T3 after six months of treatment interruption.

**Table 1 children-10-00691-t001:** Demographic characteristics of the sample.

	ESDM (n = 18)
	Means (SD) or Numbers (%)
Children’s age at baseline (months)	29.5 (6.6)
Maternal age at conception (years)	32.05 (5.1)
Paternal age at conception (years)	35.3 (5.5)
**Sex**	
Male	16 (88.8%)
Female	2 (11.1%)
**Nationality**	
Italian	12 (66.6%)
Others	6 (33.3%)
**Siblings with ASD**	1 (5.5%)

**Table 2 children-10-00691-t002:** Parents’ occupational status and educational level.

**Maternal Occupational Status**	
Employed	12 (66.6%)
Unemployed	1 (5.5%)
Other/missing	5 (27.7%)
**Paternal Occupational Status**	
Employed	17 (94.4%)
Unemployed	1 (5.5%)
Other/Missing	0
**Maternal Educational Level**	
Middle school	5 (27.7%)
High school	5 (27.7%)
University degree or higher	6 (33.3%)
Other/missing	2 (11.1%)
**Paternal Educational Level**	
Middle school	6 (33.3%)
High school	6 (33.3%)
University degree or higher	5 (27.7%)
Other/missing	1 (5.5%)

**Table 3 children-10-00691-t003:** Median scores of sample patients in T2 and T3.

	T2 Median	T3 Median	T2–T3	
			*p*	r
**ADOS-2**				
*CSS-SA*	3.5	4	0.129	0.36
*CSS-RRB*	8	7	0.044 *	0.47
*CSS Tot*	4	5	0.216	0.29
** *VABS-II* **				
*Communication*	63	71	0.097	0.39
*Daily living*	60.5	60.5	0.717	0.09
*Socialization*	67	67	0.276	0.26
*Total*	59.5	63	1	0
** *BOSCC* **				
*Social communication*	17.5	18	0.663	0.1
*Restricted and repetitive behaviors*	5	3	0.028 *	0.52
*Total*	24	21	0.396	0.2

Note: *p* values result from paired-sample Wilcoxon tests. r values represent the effect size of each test. Asterisk indicates statistical significance.

**Table 4 children-10-00691-t004:** Mean and range score in T2 and T3.

	T2		T3	
	Mean	Range Score	Mean	Range Score
**ADOS-2**				
*CSS-SA*	3.89	1–7	4.65	1–9
*CSS-RRB*	7.67	4–10	6.76	1–9
*CSS Tot*	4.56	1–7	5.18	1–9
**VABS-II**				
*Communication*	60.4	28–87	64.6	32–98
*Daily living*	64.6	47–99	63.3	45–99
*Socialization*	66.8	46–88	69.9	49–101
*Total*	62.4	40–95	62.1	35–97
**BOSCC**				
*Social communication*	19.1	5.50–37	18.3	2–36.5
*Restrictive and repetitive behaviors*	5.44	2–9.50	4.33	1–18.5
*Total*	24.5	7.50–46.5	22.5	3.50–43.5

Note. T2 means after 12 months of the intervention and T3 after six months of treatment interruption.

## Data Availability

Data will be made available upon reasonable request.

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
