# Peer review of "Short-Term Outcomes of an ESDM Intervention in Italian Children with Autism Spectrum Disorder following the COVID-19 Lockdown"

_children, 2023, doi:10.3390/children10040691_

Round 1

Reviewer 1 Report

Thank you for the opportunity to review the present paper. The topic of the paper is of great importance, and we agree that the study is novel and should be included in the literature. Overall, the paper was well written and clear. 

Lines 221-225 present results that are similar to those presented in the previous paragraph and are also presented in the following table. This should be reviewed to avoid redundancy.

Author Response

Response to Reviewers

Many thanks for your work. We are glad to know that overall, our work was favorably received. Please, find below our point-by-point to each of your comments.

We emphasized that the aim of the study was to demonstrate that children who had received a year of continuous ESDM treatment in person did not experience any worsening during the lockdown imposed by the Covid19 pandemic. We also incorporated some information related to previous studies reported in the introduction and modified some parts of the text that, as suggested by the reviewers, could be better placed in specific paragraphs. Additionally, we made revisions to the language and style. Finally, we provided a graphical abstract to help the reader better understand the flow of the study's work.

We thank you for the thoroughness with which our work has been evaluated.

Response to Reviewer 1

  • Lines 221-225 present results that are similar to those presented in the previous paragraph and are also presented in the following table. This should be reviewed to avoid redundancy.

Thank you for your comment. We have made the necessary changes to the paragraph.

Reviewer 2 Report

Thank you for the opportunity to review this manuscript.

A brief summary 

This paper reports on maintenance of treatment effects for 18 children who received one-year of low-intensity Early Start Denver Model (ESDM) therapy prior to a 6-month government-mandated lockdown period. During the lockdown period, parents received online instruction and support in implementing the ESDM with their child, however the nature of this remote support was not consistent across participants. A range of child outcomes were measured at the end of the 6-month lockdown period (T3) and compared with data taken at the end of the one-year direct intervenion period (T2). Results indicate that overall, children maintained gains in social communication skills and demonstrated a decrease in restricted and repetitive behaviours.

This is an interesting and important study, as many autistic children and families were unable to access direct support/intervention for periods during the covid pandemic. As the authors note, it is encouraging that participating children appeared to maintain gains observed during intervention. However, due methodological issues it is not possible to attribute this maintenance to the ESDM intervention that participants received prior to the lockdown.

General concept comments
I would suggest that the authors consider following the language preferences of the autistic community (see Monk et al., 2022).

Specific comments 

Line 53-54 – The authors state that the 2020 Fuller et al. meta-analysis reported on 640 children who received the ESDM. However, according to Fuller et al. only 286 children received the intervention, the remaining 354 were in control groups.

Line 56 – “moderate to significant” should read “moderate significant.” It would be helpful to also state that there were no significant effect sizes observed for other measures and there were concerns regarding the scientific rigor of many of the included studies.

Lines 63 – 65 – It is unclear how the statement regarding differences between Asian and Western countries relates to the present study.

Lines 74 – 76 – What was the intervention dosage in the Contaldo et al. study?

Lines 89 – 90 – Please specify what “very low intensity” means. How many hours per week?

Line 107 – It’s unclear what is meant by the “initial lockdown,” was there more than one lockdown related to the present study?

Lines 135 – 140 – More detail is needed regarding the parent-mediated intervention. How often did parents receive “remote instruction and support”? The authors note that “not all parents could put into practice…” what data were collected regarding parents’ use of the techniques in order to reach this conclusion?

Discussion section – The authors state that the purpose of the study was to evaluate maintenance after discontinuation of an intervention, but it reads as though they are evaluating a remote parent-mediated intervention. This is somewhat confusing for the reader.

Line 246 – I would be interested to hear the authors thoughts on why a reduction in RRBs was observed from T2 – T3, but not from T1 – T2. This is perhaps the most interesting finding, in part because the ESDM does not directly target RRBs. but the authors do not discuss the finding or its implications.

Lines 255 – 267 – It would be helpful to provide further detail on the cited studies (e.g., mode and dosage of intervention) and explicitly show how previous findings might relate to the present study.

Lines 277 – 282 – This detail may be better suited to the Methods section.

Lines 288 – 290 – What evidence is there to suggest that the pre-pandemic treatment period made parents more receptive to therapists’ advice during the lockdown?

Line 292 - Siracusano et al. is listed as reference number 34 in the reference list and number 35 in text.

Lines 296 – 300 – The authors suggest that the unique conditions of the lockdown period (e.g., increased presence of parents at home) may have contributed to the reduced severity of maladaptive behaviors. However, the present study did not report on maladaptive behaviors so it is unclear what the authors are referring to.

Lines 308 – 310 – The authors do not appear to have collected any data related to the quality or quantity of parent-child interactions to substantiate this claim.

Lines 342 – 343 – Were all participating children capable of providing informed consent or did parents provide consent on their behalf? What attempts were made to seek child assent to participate?

References:

Monk, R., Whitehouse, A. J., & Waddington, H. (2022). The use of language in autism research. Trends in Neurosciences. https://doi.org/10.1016/j.tins.2022.08.009

Author Response

Response to Reviewers

Many thanks for your work. We are glad to know that overall, our work was favorably received. Please, find below our point-by-point to each of your comments. Specific reviewer’s comments are reported in black and our responses in blue.

We emphasized that the aim of the study was to demonstrate that children who had received a year of continuous ESDM treatment in person did not experience any worsening during the lockdown imposed by the Covid19 pandemic. We also incorporated some information related to previous studies reported in the introduction and modified some parts of the text that, as suggested by the reviewers, could be better placed in specific paragraphs. Additionally, we made revisions to the language and style. Finally, we provided a graphical abstract to help the reader better understand the flow of the study's work.

We thank you for the thoroughness with which our work has been evaluated.

Response to Reviewer 2

  • I would suggest that the authors consider following the language preferences of the autistic community (see Monk et al., 2022).

Thank you for the comment. We used the bibliography given and modified with the language preferences of the autism community.

  • Line 53-54 – The authors state that the 2020 Fuller et al. meta-analysis reported on 640 children who received the ESDM. However, according to Fuller et al. only 286 children received the intervention, the remaining 354 were in control groups.

Thank you for the comment. We verified and edited by specifying the number of children who had received intervention and controls. (Line 54)

  • Line 56 – “moderate to significant” should read “moderate significant.” It would be helpful to also state that there were no significant effect sizes observed for other measures and there were concerns regarding the scientific rigor of many of the included studies.

Thank you for the comment. We have corrected the phrase "moderate to significant" and highlighted the points mentioned by the reviewer.

  • Lines 63 – 65 – It is unclear how the statement regarding differences between Asian and Western countries relates to the present study.

We decided to also report the work of Wang et al. in support of the effectiveness of ESDM. it was not our intention to emphasize any specific difference between Asian and Western countries.

  • Lines 74 – 76 – What was the intervention dosage in the Contaldo et al. study?

We have described in detail the type of intervention and intensity presented in Contaldo et al. (line 79-82).

  • Lines 89 – 90 – Please specify what “very low intensity” means. How many hours per week?

We have added in the text: “2 hours per week” (line 96)

  • Line 107 – It’s unclear what is meant by the “initial lockdown,” was there more than one lockdown related to the present study?

Thank you for the note. We have revised the statement to clarify the author Degli Espinosa's intended meaning. We assume that the author was referring to the first and most severe lockdown in Italy, as there have been subsequent lockdowns that are not comparable to the first one in March 2020, which halted all interventions.

  • Lines 135 – 140 – More detail is needed regarding the parent-mediated intervention. How often did parents receive “remote instruction and support”? The authors note that “not all parents could put into practice…” what data were collected regarding parents’ use of the techniques in order to reach this conclusion?

We specified in the text that therapists followed families and children remotely for average two hours a week. We did not collect data to show that some parents were unable to implement the indications, but this observation refers to the feedback the therapists collected in their relationship with the parents.

  • Discussion section – The authors state that the purpose of the study was to evaluate maintenance after discontinuation of an intervention, but it reads as though they are evaluating a remote parent-mediated intervention. This is somewhat confusing for the reader.

It was not our first aim to evaluate a remote parent-mediated intervention, but we believe it could be a point of strength to highlight in the text. For this reason, we included this issue in the discussion.

  • Line 246 – I would be interested to hear the authors thoughts on why a reduction in RRBs was observed from T2 – T3, but not from T1 – T2. This is perhaps the most interesting finding, in part because the ESDM does not directly target RRBs. but the authors do not discuss the finding or its implications.

We are impressed by this result as well, particularly considering that we did not find the same in the previous study. However, in the present study, two measures confirm this data, both ADOS and BOSCC. We are aware that there is poor evidence in the literature about RRBs, as also reported by Berry (2019) and it is a field that worths further exploration. We have integrated this part into the discussion by adding some possible speculations (line 281-302)

  • Lines 255 – 267 – It would be helpful to provide further detail on the cited studies (e.g., mode and dosage of intervention) and explicitly show how previous findings might relate to the present study.

In response to your request for further detail on the cited studies and their relation to the present study, we have provided additional information on the mode and dosage of the interventions from the studies by Rogers et al. [31] and Estes et al. [32], from line 268 to line 280.

  • Lines 277 – 282 – This detail may be better suited to the Methods section.

We have moved one part to the section on Study Design. However, we also believe the remaining part is consistent within the limitations of our study, as parents did not receive fidelity from therapists, as reported by Rogers and Dawson (P-ESDM).

  • Lines 288 – 290 – What evidence is there to suggest that the pre-pandemic treatment period made parents more receptive to therapists’ advice during the lockdown?

This is no clear evidence about it, but rather it is our belief based on the direct experience that therapists have with the families involved in in-person intervention sessions, who share with the therapists the treatment’s objectives for implementation ad home. We have modified in the text by inserting the sentence "we strongly believe that" in order to clarify that this is not evidence but rather an assumption.

  • Line 292 - Siracusano et al. is listed as reference number 34 in the reference list and number 35 in text.

Thanks, we have modified.

  • Lines 296 – 300 – The authors suggest that the unique conditions of the lockdown period (e.g., increased presence of parents at home) may have contributed to the reduced severity of maladaptive behaviors. However, the present study did not report on maladaptive behaviors, so it is unclear what the authors are referring to.

Perhaps the terminology is prone to confusion. We used the word "maladaptive", as reported by Fulton (2014), meaning "disruptive, destructive, aggressive or significantly repetitive behaviors". We have therefore modified the terminology in the text so as not to make the message confusing.

  • Lines 308 – 310 – The authors do not appear to have collected any data related to the quality or quantity of parent-child interactions to substantiate this claim.

We adopted as measure the BOSCC, which specifically detect the quantity and quality of parent-child interactions. In addition, we report Rogers' findings as data supporting the importance of direct parental involvement.

  • Lines 342 – 343 – Were all participating children capable of providing informed consent or did parents provide consent on their behalf? What attempts were made to seek child assent to participate?

Informed consents were collected from all parents of the children involved in the study. We have modified the sentence, thank you.

Reviewer 3 Report

This is a well-done paper that addresses an important and timely issue.  The introduction provides a good overview of the existing literature with a more focused discussion of studies from Italy. In the early section of the introduction which reviews meta-analyses, it might be helpful to note the intensity of the intervention efforts provided, since this is quite variable and may influence outcomes. It might also be helpful to provide some explanation for the conclusion that ESDM is viewed as more effective than other methods (lines 84-85). The authors provide a clear overview of the project which preceded this research and a clear statement of their aims in this project.

Similarly, the methods section offers a clear statement of the research plan including the reason for the lack of a control group.  Inclusion and exclusion criteria are clear and the statistical analyses are appropriate.  The demographic characteristics might be condensed into one table instead of three.  Several aspects of the research should be further detailed. The authors note that services during the pandemic were highly variable; more information about those services would be helpful. In addition, it is unclear why the authors did not administer the ESDM checklist after the services provided during the pandemic. That might have provided quantitative data about possible areas of development which were more affected by the transition to remote services.

The results are clearly presented but it may have been helpful to include the mean and range of scores for each measure. That would allow the reader to more fully ascertain the variability in the population and its changes over time.

The discussion appears to be the section of the paper where revisions might be most helpful. First, some of the content of the discussion, including the continued discussion of previous studies might fit better in the Introduction. The discussion section could then be limited to the ways in which the current findings support or refute previous work. Similarly, the description of parent coaching included in the discussion of limitations, might be better placed in the methods section. The finding that the parent-mediated intervention was associated with a reduction in RRBs is an important one that merits further discussion here.  How do the authors explain this finding? Similarly, the assertion in the last paragraph that parent involvement is critical to maintaining child progress should be further developed.

There are several related areas that might be considered in the discussion. First, the fact that some families may not be able to implement or benefit from parent mediated intervention should be explored further.  This is a potential limit to the generalizability of the findings and merits further discussion. The authors note this briefly in the methods section; it could be discussed more fully in the discussion. Secondly, while the data suggest that children may maintain previous goals, they also suggest that children will not make additional progress or that such progress is limited to specific areas of development. This should be noted. Finally, the authors have an opportunity to provide recommendations for the use of telehealth family support, especially for those families who have limited access to services. Those recommendations would be a helpful addition to this article. 

There are numerous grammatical errors that affect the clarity of the writing. These are likely due to translation difficulties. A careful reading of the manuscript by a native English speaker is recommended to address these. 

This paper makes a valuable contribution to the literature on the effectiveness of parent-mediated interventions such as ESDM. The authors used a naturally occurring disruption to services to assess the impact of diminished access to professional intervention on child outcomes. The paper is well written and the results are clearly described. The paper will be of considerable value to clinicians and researchers alike.

Author Response

Response to Reviewers

Many thanks for your work. We are glad to know that overall, our work was favorably received. Please, find below our point-by-point to each of your comments. Specific reviewer’s comments are reported in black and our responses in blue.

We emphasized that the aim of the study was to demonstrate that children who had received a year of continuous ESDM treatment in person did not experience any worsening during the lockdown imposed by the Covid19 pandemic. We also incorporated some information related to previous studies reported in the introduction and modified some parts of the text that, as suggested by the reviewers, could be better placed in specific paragraphs. Additionally, we made revisions to the language and style. Finally, we provided a graphical abstract to help the reader better understand the flow of the study's work.

We thank you for the thoroughness with which our work has been evaluated.

Response to Reviewer 3

  • It might also be helpful to provide some explanation for the conclusion that ESDM is viewed as more effective than other methods.

We have reported the conclusion of Cucinotta et al. who investigated the impact of three different types of early interventions on the developmental profile of 90 toddlers with ASD divided in three groups, including 36 children who received treatment as usual, 13 children who received an intervention based on early intensive behavioral intervention (EIBI) and 41 children who received the ESDM. The results of the study showed that all three interventions improved the developmental profile of the participants, but the authors emphasize the effectiveness of intervention based on naturalistic developmental behavioral principles, such as the Early Start Denver Model. We have tried to specify better this issue in the text.

  • The demographic characteristics might be condensed into one table instead of three.

Unfortunately, we had to split the sociodemographic table to comply with the journal's criteria for the number of tables. In any case, we have merged the tables related to occupation and education level, in order to make the reading of the reported socio-demographic data clearer.

  • More information about those services would be helpful.

Our outpatient public services deal with all the neuropsychiatric disorders in childhood and adolescence, not just neurodevelopmental disorders and several professionals work there, including psychologists, nurses, speech therapists, psychiatric rehabilitation technicians, and physiotherapists. An important aspect to highlight, which has been mentioned in both this article and the previous one, is that the therapists we refer to have received specific training on ESDM and obtained fidelity from a certified trainer according to the procedures reported in Dawson and Rogers. For this reason, we are not sure it is essential to provide a detailed description of the organization of the services, because we fear that it might be confusing for the reader.

  • It is unclear why the authors did not administer the ESDM checklist after the services provided during the pandemic.

We decided not to administer the ESDM checklist because the children were not undergoing a standardized and manualized intervention during the period of lockdown caused by the pandemic, whereas instead the therapists provided remote support to families without following the standardized procedure of P-ESDM, as highlighted in the text.

  • The results are clearly presented but it may have been helpful to include the mean and range of scores for each measure.

Thank you, as suggested we have added a table that reports the means and score ranges of the tests used.

  • The description of parent coaching included in the discussion of limitations, might be better placed in the methods section.

We would like to clarify that what has been done with parents is described in the methodology section, as we have reported the mode of remote support provided by professionals during the lockdown period. The information provided in the discussion, specifically within the limitations of the study, is intended to reiterate some of the fundamental concepts.

  • The finding that the parent-mediated intervention was associated with a reduction in RRBs is an important one that merits further discussion here. 

In this study, two measures confirm this data, both the ADOS and the BOSCC. We are aware that this is a topic poorly described in the literature, as also reported by Berry (2019). As a result, we have integrated the text with our own interpretation and possible explanation (line 281-302). It is certainly a field that needs further exploration.

  • The assertion in the last paragraph that parent involvement is critical to maintaining child progress should be further developed.

In response to the reviewer's comment regarding the development of the assertion that parent involvement is critical to maintaining child progress, we have expanded on this aspect in the revised text. We have emphasized the importance of parental involvement in the intervention process and its potential for contributing to the child's skill acquisition, generalization, and overall success of the intervention approach. (line 348)

  • The fact that some families may not be able to implement or benefit from parent mediated intervention should be explored further.  This is a potential limit to the generalizability of the findings and merits further discussion.

We agree with the reviewer's comment and have indeed incorporated this aspect into the limitations of our study. Some families, given the specific circumstances related to the pandemic (e.g., being confined indoors without access to outdoor or large spaces, having to work remotely, having multiple children in the home simultaneously), were unable to implement all the guidance provided by professionals. However, thanks to the weekly support, they were still able to have a direct and useful exchange with the professionals. Despite this, we believe that the direct involvement of families is crucial for promoting the generalization.

  • While the data suggest that children may maintain previous goals, they also suggest that children will not make additional progress or that such progress is limited to specific areas of development. This should be noted.

Thank you, we have further highlighted this aspect in the conclusion section.

  • The authors have an opportunity to provide recommendations for the use of telehealth family support, especially for those families who have limited access to services. Those recommendations would be a helpful addition to this article.

Thank you for the note. We have integrated in the conclusion section, emphasizing the importance of offering different opportunities of intervention modalities that families can access.

  • There are numerous grammatical errors that affect the clarity of the writing. These are likely due to translation difficulties. A careful reading of the manuscript by a native English speaker is recommended to address these.

Thank you, we have asked a native English speaker to proofread the article.

Round 2

Reviewer 2 Report

Thank you for the opportunity to review a revised version of this manuscript.